# Women's experiences of and satisfaction with childbirth: Development and validation of a measurement scale for low- and middle-income countries

Meghan A. Bohren[1]*, Charbel Abi Saad[2], Charles Kabore[3], Kristi Sidney Annerstedt[4], Claudia Hanson[4,5], Myriam de Loenzien[2], Simon Tiendrebeogo[3], Fadima Bocoum[3], Marion Ravit[2], Camille Etcheverry[2], Pisake Lumbiganon[6], Nampet Jampathong[6], Guillermo Carroli[7], Celina Gialdini[7], Quoc Nhu Hung Mac[8], Helle Mölsted Alvesson[4], Andrainolo Ravalihasy[2], Alexandre Dumont[2], Ana Pilar Betrán[9]

1 Gender and Women's Health Unit, Nossal Institute for Global Health, School of Population and Global Health, University of Melbourne, Carlton, Victoria, Australia, 2 Université Paris Cité, IRD, Inserm, Ceped, Paris, France, 3 Institut de Recherche en Sciences de la Santé, Ouagadougou, Burkina Faso, 4 Department of Global Public Health, Karolinska Institutet, Stockholm, Sweden, 5 Department of Disease Control, Faculty of Infectious and Tropical Diseases, London School of Hygiene and Tropical Medicine, London, United Kingdom, 6 Department of Obstetrics and Gynaecology, Faculty of Medicine, Khon Kaen University, Khon Kaen, Thailand, 7 Centro Rosarino de Estudios Perinatales, Rosario, Argentina, 8 Pham Ngoc Thach University, Ho Chi Minh City, Vietnam, 9 UNDP/UNFPA/UNICEF/WHO/World Bank Special Program of Research, Development and Research Training in Human Reproduction (HRP), Department of Sexual and Reproductive Health and Research, World Health Organization, Geneva, Switzerland

* meghan.bohren@unimelb.edu.au

## Abstract

### Background

Measuring person-centered maternity care outcomes typically consists of two types of measures: experiences of care and satisfaction with care. There are limited validated measurement tools for these measures, particularly in low- and middle-income countries (LMICs). The QUALI-DEC study aims to improve decision-making around caesarean section. We describe development of the QUALI-DEC Study Birth Experience and Satisfaction (QD-BES) scale, and scale validation in Argentina, Burkina Faso, Thailand, and Viet Nam.

### Methods

We used a three-phase scale development and validation approach: 1) item development, 2) scale development, and 3) scale evaluation. We systematically identified existing tools, and assessed them using the QUALI-DEC theory of change, study context, and psychometric qualities. We proposed the 10-item QD-BES scale to balance feasibility, theoretical coverage, and comprehensiveness. We conducted a

**Data availability statement:** The datasets used and/or analyzed during the current study will be available from the completion of the QUALI-DEC trial in July 2025 via the QUALI-DEC study Zenodo platform: https://zenodo.org/communities/qualidec/records?q=&l=list&p=1&s=10&sort=newest. Requests regarding data use should be directed to the communications officer for the project, Ouarda Lunetta-Namane (qualidecproject@gmail.com).

**Funding:** This study is part of the QUALI-DEC-project which is co-funded by the European Union's Horizon 2020 research and innovation programme under grant agreement No 847567 and by UNDP-UNFPA-UNICEF-WHO-World Bank Special Programme of Research, Development and Research Training in Human Reproduction (HRP), a cosponsored programme executed by the World Health Organization (WHO) in the Department of Sexual and Reproductive Health and Research (SRH). Alexandre Dumont was awarded the Horizon Birth Day Prize from the European Commission in 2018, which contributed to the development of this project. MAB is supported by an Australian Research Council Discovery Early Career Researcher Award (DE200100264) and a Dame Kate Campbell Fellowship (University of Melbourne Faculty of Medicine, Dentistry, and Health Sciences). The funders had no involvement in the study design, data collection, analysis, interpretation of data, writing the article, or the decision to submit for publication. The contents of this article are solely the responsibility of the authors and do not reflect the views of the EU, UNDP, UNFPA, UNICEF, WHO, or the World Bank or their respective institutions.

**Competing interests:** The authors have declared that no competing interests exist.

baseline exit survey with post-partum women in 32 hospitals in 4 countries. We conducted exploratory factor analysis (EFA), and confirmatory factor analysis (CFA).

## Results

3127 women participated, most were multiparous (61.0%), without previous caesarean section (77.2%), and preferred vaginal birth (72.8%) despite high rates of caesarean section (39.4%). EFA identified three dimensions: emotional satisfaction (3-items), support and respect by providers (4-items), and communication with providers (3-items), with high loading coefficients (0.5–0.97). CFA confirmed the three-dimension scale, with good model fit (CFI and IFI: 0.95, Cronbach's alpha: 0.70–0.90). Criterion validity was assessed by exploring characteristics of women, obstetric histories, and birth experiences.

## Conclusions

We present psychometric validation of a scale measuring women's satisfaction with care and experiences of childbirth care, using a systematic approach to development and validation in four LMICs. The 10-item QD-BES-scale is short, easily-administered, valid, and reliable. The QD-BES-scale is useful to contribute to the generation of new knowledge about quality of maternity care in LMICs, as well as help to meet the major challenge of implementing and measuring respectful care at scale.

## Introduction

Since 2018, three landmark reports [1–3] have highlighted the importance improving quality of care on the pathway to achieving effective universal health coverage under Sustainable Development Goal 3 [4]. These reports define quality of care as care that maintains or improves health and is person-centered, meaning that it is "respectful of and responsive to individual preferences, needs and values" [1,5]. Providing person-centered care is important both because people have a right to be treated with dignity and respect in healthcare settings, and because person-centered care is associated with improved health outcomes and use of healthcare services [6,7]. The World Health Organization (WHO) framework for quality of care for pregnant women and newborns situates quality of care across the provision of care (evidence-based practices for routine and management of complications, actionable information systems, and functional referral systems) and the experience of care (effective communication, respect and dignity, and emotional support), which are underpinned by competent and motivated human resources and essential physical resources [8].

In theory, measuring person-centered outcomes allows for both quality improvement processes to be evaluated and for health services and health systems to be held accountable to the communities that they serve. However, in reality, person-centered outcomes are difficult to measure and interpret, as they rely on a person's report of their healthcare encounter and are affected by limited precision

and clarity in how the indicator was developed [7]. The two main categories of person-centered measures of quality of care are 'experiences of care' and 'satisfaction with care', which are related concepts with some distinct differences [7]. Experiences of care reflect the interpersonal aspects of quality care, typically communication with health workers, respect, and emotional support. Experience of care measures are process measures, which means that it measures the extent to which the person had "good quality care" [7]. Process measures are typically more sensitive to differences in quality of care compared to outcome measures, and can sometimes be easier to interpret and make concrete actions to improve. In comparison, satisfaction with care assesses whether the care provided has met the person's needs and expectations [7]. A person's needs and expectations may evolve throughout a healthcare encounter, depending on the way care is provided. Satisfaction with care measures are outcome measures, which means that it is an endpoint or measure of effect of the healthcare interaction (or intervention) [7]. Satisfaction measures are useful for identifying areas of service provision that are important to individuals, or at an aggregate level for communities. It is important to note that a person's experience of care and health outcomes can directly impact their satisfaction with care, and also indirectly, by influencing their needs, priorities, and values, which in turn affect satisfaction [7].

While experience of and satisfaction with care measures are critically important to understanding quality of care during pregnancy and childbirth, there is limited research conducted in this space, few validated measurement scales, and even fewer measurement scales validated or developed in low- and middle-income country (LMIC) settings. A recent scoping review by Larson and colleagues aimed to identify measures and instruments for assessing experiences of care in any setting globally [9]. Of 171 articles included, half did not use a validated instrument, and most (>60%) were conducted in upper-middle or high-income countries [9]. Most studies included only one or two domains of user experience (most commonly communication and respect) and no studies evaluated user experience across all domains (communication, respect and dignity, emotional support, user-centeredness) [9]. Similarly, a systematic review of only validated instruments for measuring women's childbirth experiences and satisfaction by Nilvér and colleagues included 46 papers with 36 instruments, and only two studies were conducted in LMICs (Jordan, Senegal); the remaining studies were all conducted in North America and Europe [10]. Many of the included scales measured very specific aspects of childbirth or within specific populations, such as women with traumatic childbirth or preterm birth, and few studies reach all expected standards for the quality of psychometric properties [10]. Moreover, tools developed and validated in high-income settings typically included items that may not be relevant to LMIC settings, for instance around water birth or one-to-one midwifery care [10]. These two reviews demonstrate the significant gap in the evidence. Substantial work is needed to develop and validate tools to measure women's experiences of and satisfaction with care, particularly in LMIC settings, and in a general population of women (rather than specific groups with complications).

## The QUALI-DEC project

Caesarean section can be a life-saving intervention for women and babies when medically indicated; however an increasing proportion are performed without clinical indication both globally and in LMIC settings [11]. As with any surgery, a caesarean birth has risks, and the overuse diverts scarce resources in LMIC settings, and consequently can reduce access to healthcare for all women [12]. Overuse of caesarean section is influenced by both clinical practices that are not evidence-based, as well as non-clinical factors, such as cultural, social and structural influences [12–15], that must be accounted for to effectively address rising caesarean section rates. The QUALI-DEC project (QUALIty DECision-making by women and providers) was designed to improve decision-making for caesarean section by women and health workers using a multi-faceted intervention in four LMICs: Argentina, Burkina Faso, Thailand and Viet Nam [16], and is based on the 2018 WHO recommendations for non-clinical interventions to reduce unnecessary caesarean section [17]. The QUALI-DEC strategy involves four main intervention components: 1) decision analysis tool for shared decision-making about birth, 2) labor and birth companionship, 3) audit and feedback about caesarean section, and 4) opinion leaders to implement best practices [16].

To assess the effectiveness of the QUALI-DEC strategy on women's experiences of and satisfaction with care and other secondary outcomes, a cross-sectional survey is planned at two time points, before and after the intervention implementation [16]. In this article, we describe the development of a scale for assessing women's birth experiences and satisfaction in the context of the QUALI-DEC project, and report the validation of the tool in four LMICs: Argentina, Burkina Faso, Thailand and Viet Nam.

## Materials and methods

We report this analysis according to the Strengthening the Reporting of Observational Studies in Epidemiology (STROBE) Statement [18]. The QUALI-DEC study protocol has been published [16] and is prospectively registered (ISRCTN67214403). In the following sections, we first describe the scale development, followed by the primary data collection and validation in Argentina, Burkina Faso, Thailand and Viet Nam. We used a three-phased data approach to develop and validate the QUALI-DEC Birth Experience and Satisfaction (QD-BES) scale [19]: 1) item development, 2) scale development, and 3) scale evaluation.

### Phase 1. Item development

Given the challenges and limitations described above related to validated measures of women's experiences of and satisfaction with care, and our previous experience working in this area, we started with an informed and pragmatic approach to scale development. First, we reviewed relevant scientific literature and related measurement efforts in maternal health. Namely, this included reviewing the studies included in the Larson 2020 scoping review [9] and the Nilver 2017 systematic review [10]. In this step, we aimed to identify previously used or proposed scales or items used globally, but with a special focus on those used in LMIC settings that may be relevant to QUALI-DEC. We also reviewed Kabakian-Khasholian's (2018) implementation study on labor companionship in Egypt, Lebanon, and Syria to assess the measures used for women's experience and satisfaction with labor companionship [20]. This study was the closest implementation study to QUALI-DEC, in terms of alignment with adapting and implementing labor companionship in three LMIC settings [20].

Through discussion and consensus with the QUALI-DEC research team (APB, CH, KSA, HMA. MR, MdL, MAB – a team with expertise in social sciences, obstetrics and gynecology, public health, demography, implementation science, and anthropology), we shortlisted the items and scales identified from the literature [9,10,20] that were relevant to the QUALI-DEC study contexts, with good psychometric properties (such as reliability and validity). We discussed where these items fit (or did not fit) given the multi-faceted intervention of QUALI-DEC, and pragmatically considering that the experience and satisfaction survey module is one of nine modules included in a survey conducted in health facilities in the immediate postpartum period (within approximately 48 hours after birth, so brevity was critical). The other survey modules relate to: 1) sociodemographic and obstetric characteristics, 2) birth outcomes, 3) knowledge about mode of birth, 4) decision-analysis, 5) labor companionship, 6) birth experience and satisfaction, 7) gender and social equity, 8) wealth index, and 9) out-of-pocket expenses. We shortlisted scales and items from three studies: Hodnett's labor agentry scale to measure control during childbirth [21], the Mackay Childbirth Satisfaction Rating Scale [22], and the WHO mistreatment of women during childbirth study [23] (Fig 1). The labor agentry scale [21] is a 29-item scale using Likert-responses, but we concluded that most scale items were not likely to be on the QUALI-DEC theory of change pathways [16] and thus we should not measure items that are unlikely to be influenced by the QUALI-DEC intervention. The Mackay childbirth satisfaction scale [22] is a 34-item scale using Likert-responses, and several (but not all) items are on the QUALI-DEC theory of change pathway. Importantly, no items measured women's birth experiences; thus the Mackay childbirth satisfaction scale would be insufficient on its own. Lastly, the WHO mistreatment study [23] includes many items relevant to women's experiences of care, as well as a 13-item set related to women's experiences of and satisfaction with care using Likert-responses. Although these 13-items were not validated as a separate scale, face and content validity testing of the

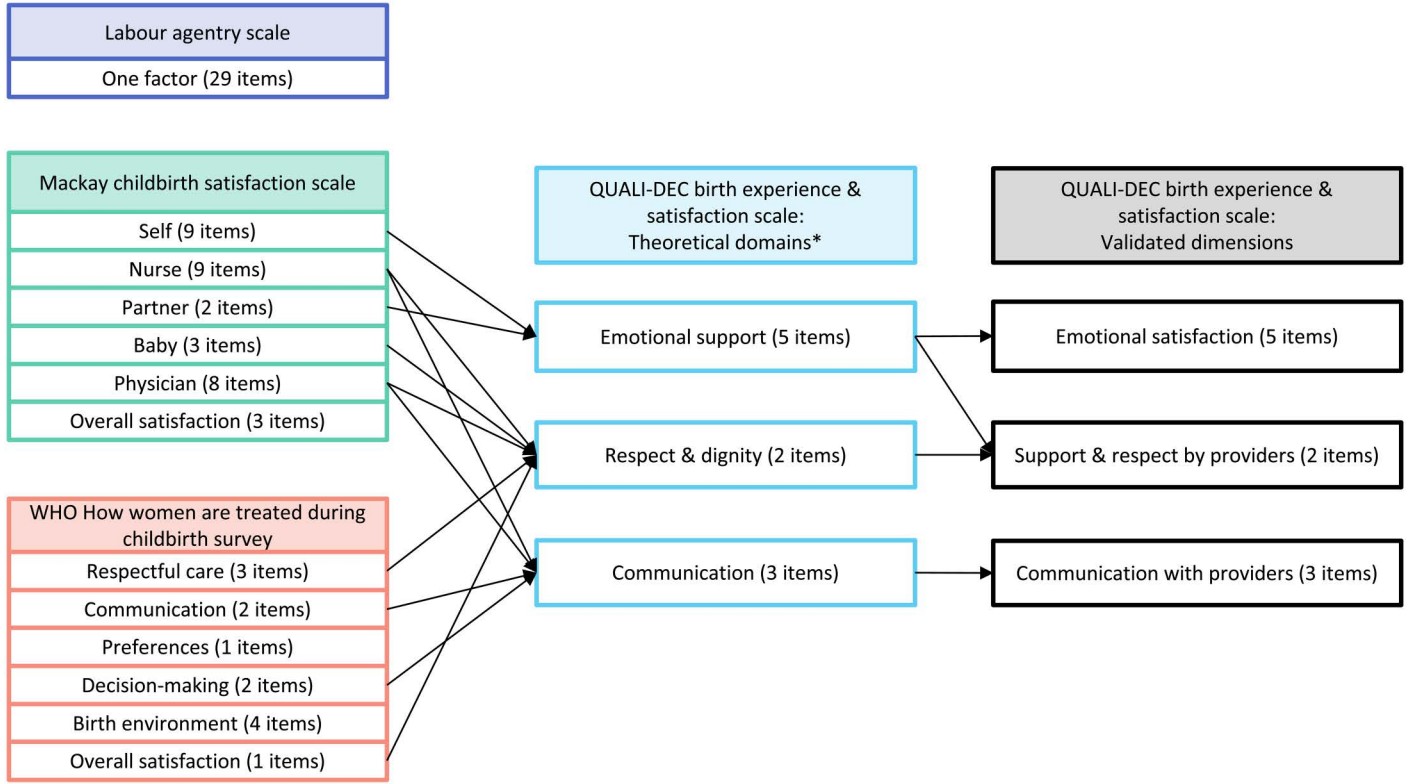

**Fig 1. Mapping and developing the QUALI-DEC birth experience and satisfaction scale.** *This figure depicts the three shortlisted tools and domains, how they map to the theoretical domains from the WHO Quality of Care Framework, and then onto the final validated dimensions.* * Based on the WHO Quality of Care Framework for Pregnant Women and Newborns [8].

entire survey was conducted in four LMICs (Ghana, Guinea, Myanmar, Nigeria) [24] and the items reflect respectful care, communication, preferences, decision-making, birth environment and satisfaction.

After selecting relevant items from the shortlisted scales, we refined and iterated the QD-BES items in two steps. First, with the QUALI-DEC international and multidisciplinary research team, we shortlisted specific items to ensure that we only measure what is included in the theory of change pathway for QUALI-DEC and what would be understandable to women across the four QUALI-DEC study countries. This process took place over several months through iterative rounds of weekly meetings and discussion to increase understanding. This process also aimed to reconcile the ultimate objectives of QUALI-DEC and of the woman's postpartum survey evaluation, with the feasibility needed for implementing the survey (e.g., the length of the survey and topics covered). Second, in each of the four QUALI-DEC countries, the refined set of items was shared and discussed with the country principal investigators and social science teams. We discussed inter-pretability, translations, and relevance for each item individually, and further refined to reflect local contexts and needs. These discussions were also informed by the initial findings from the QUALI-DEC formative research [5,16,25–31], which consisted of qualitative interviews with women, potential labor companions, and healthcare providers, as well as a health facility readiness assessment and policy document review.

## Phase 2. Scale development

The final set of questions in the QD-BES module included 10 items with 4-point Likert responses (1: strongly agree; 2: agree; 3: disagree; 4: strongly disagree), which strikes a balance between feasibility and comprehensiveness. Based on

the WHO Quality of Care Framework for pregnant women and newborns [8], the 10 items are organized in three main theoretical domains: emotional support, respect and dignity, and communication (Fig 1). Pre-testing of the QD-BES questionnaire was done with a subset of women in participating hospitals (see Phase 3. Scale evaluation) to minimize misunderstanding and subsequent measurement error, and allowed the research team to discuss and revise any items or instructions to maximize the likelihood of understandability by women.

**Phase 3. Scale evaluation**

**3.1. Study context and sites.** Box 1 describes the maternal health context in each of the four study countries. The QUALI-DEC cross-sectional survey is conducted in eight hospitals in each of the four study countries (32 hospitals total) and will be administered at two time points: baseline and after implementation. Hospital sites with relatively high caesarean section rates were purposively selected for the QUALI-DEC project with country investigators and stakeholders to reflect the range of contexts of care in each country. Of 32 hospitals, 30 were public hospitals (two private hospitals in Viet Nam), with a mix of referral level: primary (n = 1), secondary (n = 16), and tertiary (n = 15). Average annual births (2020) at the study hospitals in each country ranged from 1,528 (Argentina), 3,893 (Burkina Faso), 4,045 (Thailand), and 10,641 (Viet Nam). The caesarean section rate in the hospitals ranged from 15% to 82% in 2020 during baseline. The QUALI-DEC baseline survey with postpartum women was conducted in Argentina from 15 Dec 2021–24 June 2022, Burkina Faso from 8–26 December 2020, Thailand from 6 March 2021–3 January 2022, and Viet Nam from 8–21 October 2021.

**Box 1. Maternal health context in four QUALI-DEC study countries.**

**Argentina**
Argentina is a middle-income country in South America. The maternal mortality ratio for 2020 was estimated at 45 deaths per 100,000 live births, and >99% of births were attended by skilled birth personnel [32]. Caesarean section rates ranged between 27% and 52% within the public sector in 2017, while official data from the private sector is unavailable. According to reports by the Perinatal Reporting System, from 2009 to 2017, the use of CS has increased in the public sector by 22%, from 28% to 34%, with striking rates in some provinces being close to 50%. A study published in 2011 had shown that women in Argentina preferred vaginal birth over caesarean section [33].

**Burkina Faso**
Burkina Faso is a landlocked country located in West Africa. Burkina Faso has a high fertility rate (5.1 in 2021) [34], high maternal mortality ratio (264 maternal deaths per 100,000 live births in 2020) [32], and high neonatal mortality rate (24.7 neonatal deaths per 1000 live births). Most (80%) births take place in a health facility (2015 data) [35]. The caesarean section rate from 2010–2015 in Burkina Faso was low (3.7%) at a national level [36]. However, wide inequities in the caesarean section rate exist, with caesarean section rates in the richest quintile of women seven times higher than the rates in the poorest quintile of women [11], suggesting a context of both overuse and underuse. A study in 2016 carried out in 22 referral hospitals in Burkina Faso showed that one-fourth of caesarean sections were not medically indicated and that non-medically indicated caesarean section is associated with both socioeconomic determinants and medical factors [37].

**Thailand**
Thailand is an upper-middle income country in Southeast Asia. The most recent total fertility rate and crude birth rate in 2022 were 1.0 and 5.8 respectively [38]. Maternal mortality ratio of Thailand fell gradually from 48 deaths per 100,000 live births in 2001–29 deaths per 100,000 live births in 2020 [32]. Most (99.5%) births take place in a health facility in 2022 [38]. The national caesarean section rate in 2022 was 40.9%⁸. It was predicted that with the same rate of increase, the caesarean section rate of 59.1% is projected by the year 2030 [39].

**Viet Nam**
Viet Nam is a lower-middle income country located in Southeast Asia. Its fertility rate has reached and maintained replacement level (2.09 in 2019) [1] and maternal mortality ratio has rapidly decreased in recent decades to 46 maternal deaths per 100,000 live births in 2020 [32]. Almost all (96.3%) births take place in a health facility [40]. The caesarean section rate is high (34.4%) at a national level [40], with two-thirds decided before the onset of labor. Caesarean section rates in urban areas are almost twice the rate in rural areas, with similar patterns comparing the richest quintile of women to the poorest quintile of women [40]. Previous research suggests that both medical and non-medical reasons explain the context of high rates of caesarean section [41,42].

**3.2. Participant sampling.** Sample size estimation was based on the expected difference in satisfaction scores between the period before and after the intervention. The required number of women needed for a "before-after" cross-section design was a total of 470 women per country (60 women per hospital) to ensure 90% statistical power to detect an effect size of 0.3 standard deviations or greater in satisfaction scores. The sample size calculation would ensure sufficient

power for country-specific comparisons. The calculation accounted for the clustered nature of the data by hospitals with a design effect of 2. Assuming a 10% non-response rate and 10% of ineligible women, we aimed to approach 2,256 women total: 564 women per country, 71 women per hospital.

**3.3. Study procedures and recruitment.** Data collection took place daily (including weekends) in each of the eight hospitals simultaneously over a two- to three-week period until the sample size was reached. If the sample size (n = 60 per hospital) was reached before two-weeks, data collection continued until the end of two weeks. An estimated, 5–6 women needed to be surveyed daily in each hospital to reach the required sample size. According to the average number of births in each hospital, a hospital-specific randomization factor was applied for all women giving birth the day before to randomly select 10 women. All randomly selected women were identified by a data collector who assessed their eligibility using a screening form. Women giving birth to a newborn weighting more than 1000g were eligible for the survey. Women with serious health problems or who gave birth to a malformed, stillborn, or dead child before the survey were excluded. If the woman was eligible, she was approached by a social scientist to participate in the study during her stay in the postnatal ward. If she agreed to participate, the consent form was completed, and the woman was surveyed face-to-face by the social scientist using a tablet-based data collection form. After completion of the survey, there was no further contact with the woman. A data collector also extracted relevant clinical data from her medical record for the study by the data collector.

**3.4. Study instrument.** The overall study instrument for the cross-sectional survey collects information from the medical record (Form 1) and individual data through face-to-face survey administration with post-partum women (Form 2). The QD-BES is part of the form 2 questionnaire (Table 1). The instrument was developed in English, then translated to French, Spanish, Thai, and Vietnamese to pilot and for data collection (translations available in Appendix 1).

**3.5. Data management and quality assurance.** As the questionnaire was complex (branching logic and length), data were collected with a tablet, using the REDCap application [43]. All interviewers were trained in the tool and piloted it over a three-day period. After the survey form was completed on the tablet, the data were submitted daily to the REDCap server using a 3G or internet connection. Quality checks (control for outliers, missing important elements) were conducted by the QUALI-DEC research team in real-time to identify any discrepancies during the data collection period, so that they could be easily rectified. A post-hoc check was carried out using the medical records to ensure that the variables in both the survey and the records matched.

**3.6. Data analysis.** First, an exploratory factor analysis (EFA) was used to identify the underlying dimension structure of the scale that account for the covariation among the 10 items of QD-BES scale. The model was optimized through a Promax rotation. We tested the suitability of data for factor analysis using Kaiser-Meyer-Olkin test and whether the items can be summarized with a few numbers of dimensions using the Bartlett's test.

Secondly, we performed a psychometric analysis to verify the reliability of the scale. A confirmatory factor analysis (CFA) was performed to test and validate the structure of the scale identified using EFA (test of dimensionality). To investigate the model's goodness of fit, a number of statistics were used: overall model validation test [44]; comparative fit index (CFI), incremental fit index (IFI) [45]; root mean square error of approximation (RMSEA), and the standardized root mean square residual (SRMSR) [44]. The reliability assessing the good consistency of the tool was measured by calculating the Cronbach's alpha coefficient, with a value ≥ 0.70 considered acceptable [46]. We also used the intra-class correlation coefficient (ICC), values over 0.75 showing good or excellent reliability [47]. Construct validity analyses made it possible, due to the inter-item correlation matrix, to verify whether the items of the same dimension were well correlated with each other and measured the same thing.

Discriminant validity was based on the assessment of concurrent criterion-related validity. It consisted of verifying whether test scores of the different dimensions were correlated with criteria measured at the same time [48]. Validity criterion is usually based on comparison between an existing scale and the one under development. In our case, as such appropriate scales do not exist for our dimensions, for each of them we selected criteria based on hypotheses suggested

**Table 1. The QUALI-DEC birth experience and satisfaction module.**

| # | Question | WHO quality of care domain | Validated QD-BES dimension |
|---|---|---|---|
| BES1 | *Labor can be a painful and difficult time for women.* | Emotional support | Emotional satisfaction |
| | Do you agree with this statement: "I am satisfied with my ability to cope with pain during labor". | | |
| BES2 | *You may have been looking forward to holding your baby after he/she was born, and sometimes this can take some time.* | Emotional support | Emotional satisfaction |
| | Do you agree with this statement: "I am satisfied with the amount of time that passed after the birth, before I first held my baby". | | |
| BES3 | *You were probably looking forward to feeding your baby for the first time.* | Emotional support | Emotional satisfaction |
| | Do you agree with this statement: "I am satisfied with the amount of time that passed before I first fed my baby ". | | |
| BES4 | Do you agree with this statement: "I am satisfied with the amount of time the health workers spent with me during labor". | Emotional support | Support & respect by providers |
| BES5 | Do you agree with this statement: "I am satisfied with the attitude of health workers during labor and birth". | Respect & dignity | Support & respect by providers |
| BES6 | Do you agree with this statement: "During my stay at the hospital for childbirth, health workers informed me about decisions that they took regarding my care". | Communication | Communication with providers |
| BES7 | *You probably had some concerns or fears during your stay in the hospital for childbirth.* | Communication | Communication with providers |
| | Do you agree with this statement: "During my stay at the hospital for childbirth, I feel that I had the opportunity to discuss any fears or concerns I had with a health worker". | | |
| BES8 | *You may have had some preferences about how you wanted to give birth.* | Communication | Communication with providers |
| | Do you agree with this statement: "I had the opportunity to discuss my preferences or requests with a health worker during your stay in the hospital for birth". | | |
| BES9 | *Sometimes the labor room can get busy.* | Respect & dignity | Support & respect by providers |
| | Do you agree with this statement: "I feel that my privacy was respected during examinations and treatments". | | |
| BES10 | *Based your experience during labor and childbirth, would you agree or disagree with this statement*: "Overall, I felt well supported during labor and childbirth". | Emotional support | Support & respect by providers |

All questions are on a 4-point Likert scale (1: strongly agree, 2: agree, 3: disagree, 4: strongly disagree). Italicized text is part of what the interviewer reads to the woman, in order to help her understand the context of the question.

by an expert's panel (Table 3). This approach was used by Tso et al. to validate a patient satisfaction scale [49]. For each dimension, two hypotheses were formulated regarding the links between their scores and maternal characteristics. The dimension score was obtained by adding the item scores. The score was divided into quintiles, with the top quintile considered as 'very satisfied'. Chi-squared testing was used for comparisons. Data used to test the hypotheses came either from the medical record (From 1) or from the questionnaire administered in the post-partum survey (Form 2). Statistical analyses were performed using the statistical software IBM SPSS packages version 26 (SPSS Inc., Chicago, IL, United States).

**3.7. Declarations, ethics approval and consent to participate.** In accordance with the declaration of Helsinki, scientific and ethical approvals were obtained from the following entities: 1) Ethics Committee for Health Research of Burkina Faso (Decision No. 2020-3-038), 2) the Research Project Review Panel in the UNDP/UNFPA/UNICEF/WHO/World Bank Special Programme of Research, Development and Research Training in Human Reproduction (A66006) at the WHO, 3) the WHO Research Ethics Review Committee, Geneva, Switzerland, 4) the French Research Institute for Sustainable Development, 5) the Central Research Ethics Committee; CREC (Certificate Number COA-CREC002/2021)

in Thailand, 6) Department of Reproductive Health of the Ministry of Health in Vietnam, and 7) Centro Rosarino de Estudios Perinatales of Rosario, Argentina (Record Notice No. 1/20). All methods were performed in accordance with the relevant guidelines and regulations. All women were given a study participant number to assure data anonymization and provided written informed consent prior to the survey, and surveys were conducted in a private area of the hospital.

## Results

Fig 2 shows the flowchart of participant recruitment. After exclusions, 3,127 eligible women who gave birth in the 32 participating hospitals were analyzed. The mean age was 28 years, and most women had at least secondary education (80.3%), were married (57.0%) or living with a partner (36.7%) and working as housewives (31.7%) or in the informal sector (14.2%). Most women lived in urban areas (55.0%), and approximately one-quarter of participants travelled to the hospital for the birth by motorbike (22.7%), which typically took up to one hour (<30 minutes: 53.1%, 30–60 minutes: 35.1%). Participants were mostly multiparous (61.0%), and without previous caesarean section (77.2%), Most women (90.6%) had at least four antenatal care visits. The caesarean section rate for the most recent birth was 40.4% and most were intrapartum caesareans. More than three-quarters of women had a preferred mode of birth, typically vaginal birth (84.9% women preferred vaginal birth early in pregnancy and 72.8% at the end of the pregnancy).

The sample was randomly divided into two subsamples. EFA was conducted on the first subsample and CFA on the second subsample. Appendix 2 shows item scores (mean, standard deviation, median, and correlation coefficient) for the QD-BES items and dimensions.

All 3,127 observations were used, as there were no missing values. The mean of the QD-BES items ranged from $1.29 \pm 0.64$ to $2.54 \pm 0.78$. The mean of the QD-BES summary measures ranged from $17.05 \pm 2.71$ to $19.2 \pm 2.52$. Floor and ceiling effects were calculated by percentage frequency of highest and lowest scores. All items demonstrated heterogeneous floor and ceiling effects across all women (Appendix 2). Emotional satisfaction, Support and respect by providers, Communication with providers and the global QD-BES score displayed significant floor effects (3.2%) and ceiling effects (0.4%). In addition, the correlation coefficient ranged from 0.06 to 0.85 with skewness from −0.57 to 1.85 and kurtosis from −0.70 to 4.71.

### Exploration

Following the EFA, three dimensions were identified: emotional satisfaction (3 items); support and respect by providers (4 items); and communication with provides (3 items) (Fig 1). All dimensions had an eigenvalue >1 with a scree plot showing a discernible elbow from the three dimensions (Fig 3A and 3B). Among the ten items, all had relatively high loading coefficients, ranging from 0.50 to 0.97, with minimal variation across countries (Table 2). Percentage of total variance explained by the three dimensions was 67.21% (cross-country average, Table 2). A Kaiser-Meyer-Olkin measure of sampling adequacy higher than 0.5 and a statistically significant Bartlett's test (p<0.001) insured the appropriateness of the analysis.

### Validity

CFA confirmed the structure of the scale in three dimensions. The results showed good model fit, with a Comparative Fit Index of 0.92 to 0.96 across countries (good quality model), Incremental Fit Index of 0.92 to 0.97 across countries (adequate model fit), Root Mean Square Error of Approximation of 0.07 to 0.24 across countries (acceptable model fit), and Standardized Root Mean Square Residual of 0.07 across countries (acceptable model fit) (Appendix 3). Cronbach's alpha coefficient ranged from 0.64 to 0.86 and was 0.75 to 0.89 across countries for the 10 items (Table 2). The intra-class correlation coefficient had values ranging from 0.75 to 0.96 across countries (p<0.05), meaning that there is high similarity between values from the same group (e.g., women who give birth in a particular hospital are more likely to have similar satisfaction levels to each other, compared to women who gave birth at other hospitals).

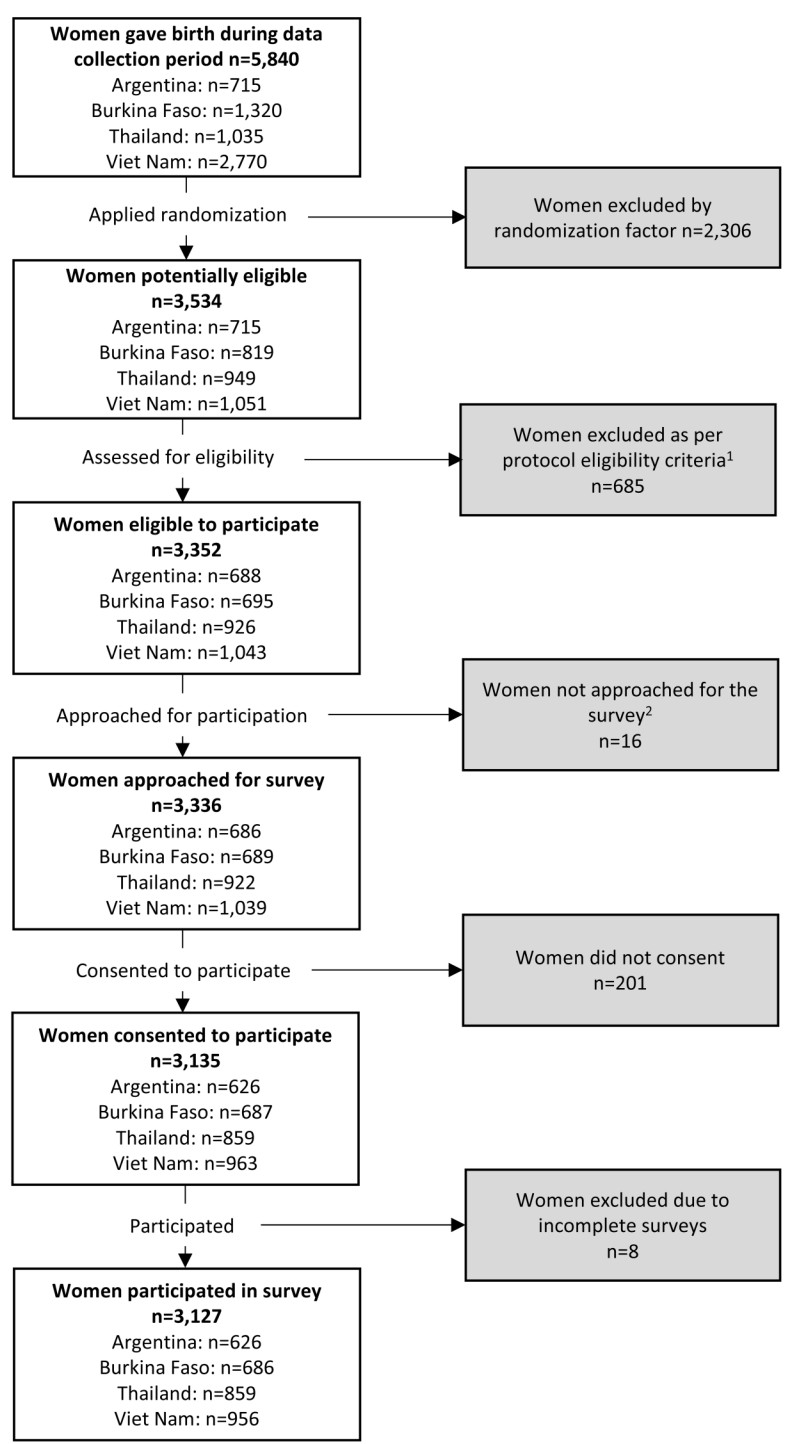

**Fig 2. Flowchart of participant recruitment.** [1]Per protocol eligibility criteria included women with no serious health concerns, or malformed, stillborn or deceased baby. [2]Typically due to discharge shortly after birth.

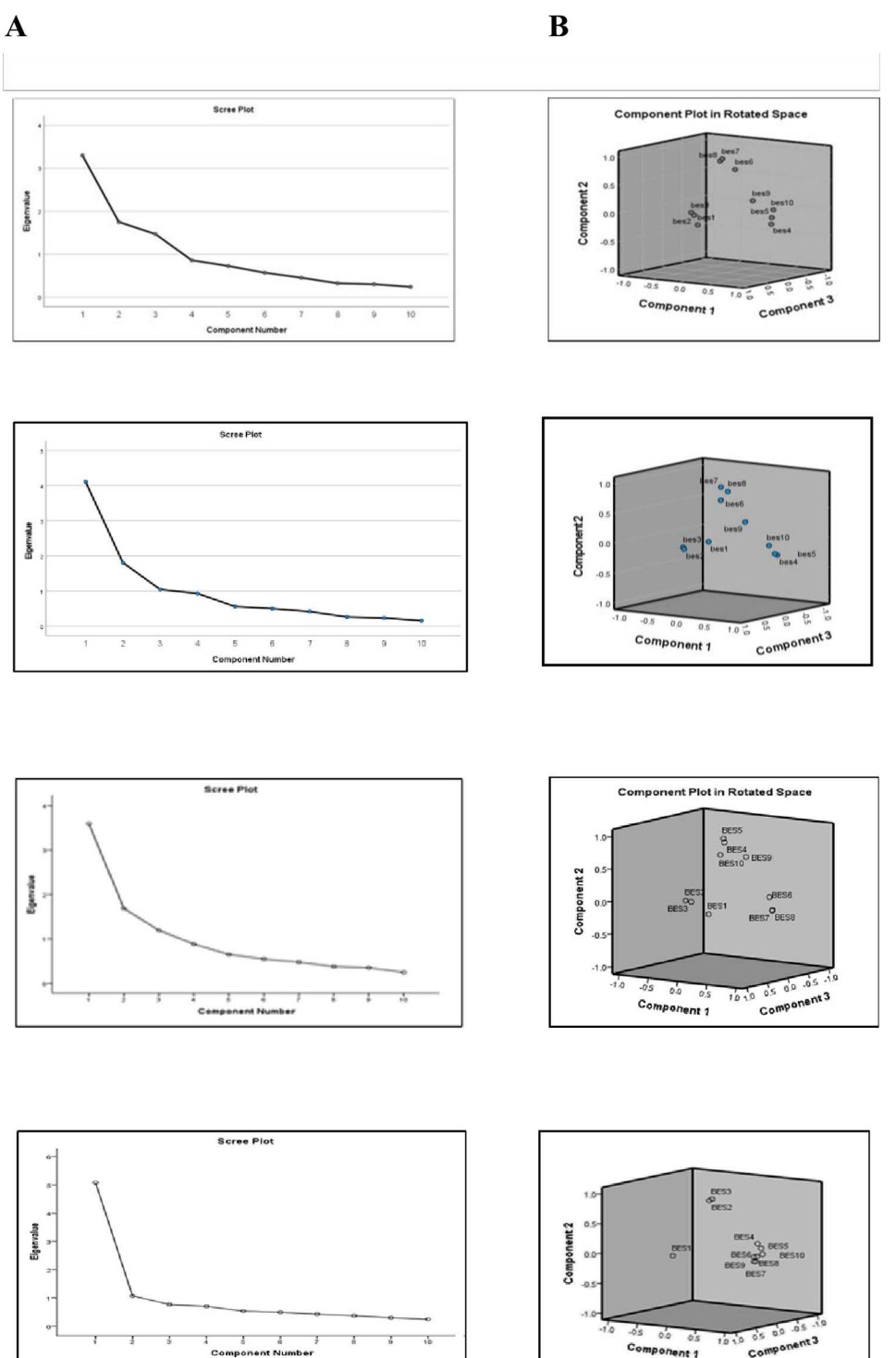

**Fig 3. Determining the number of factors to retain.Panel A depicts the scree plot for extraction of the items by principal component analysis, where the 10 BES items was taken as input variables with eigenvalues>1.** Panel B depicts a 3D schematization of the 3-factor solution.

**Table 2. QUALI-DEC birth experience and satisfaction scale dimension and items.**

| Country | Dimension | Item | Loading coefficient | Percentage of variance (cumulative) | α |
|---|---|---|---|---|---|
| **Argentina** | Emotional satisfaction | BES 1 | 0.96 | 50.77 | 0.64 |
| | | BES 2 | 0.85 | 61.50 | |
| | | BES 3 | 0.87 | 69.17 | |
| | Support & respect by providers | BES 4 | 0.71 | 76.24 | 0.83 |
| | | BES 5 | 0.83 | 81.60 | |
| | | BES 9 | 0.76 | 86.56 | |
| | | BES 10 | 0.74 | 90.83 | |
| | Communication with providers | BES 6 | 0.80 | 94.57 | 0.86 |
| | | BES 7 | 0.83 | 97.57 | |
| | | BES 8 | 0.82 | 100.00 | |
| | QD-BES | | | | 0.89 |
| **Burkina Faso** | Emotional satisfaction | BES 1 | 0.60 | 33.01 | 0.70 |
| | | BES 2 | 0.84 | 50.50 | |
| | | BES 3 | 0.85 | 65.21 | |
| | Support & respect by providers | BES 4 | 0.82 | 73.81 | 0.77 |
| | | BES 5 | 0.84 | 81.07 | |
| | | BES 9 | 0.55 | 86.77 | |
| | | BES 10 | 0.76 | 91.32 | |
| | Communication with providers | BES 6 | 0.74 | 94.56 | 0.82 |
| | | BES 7 | 0.90 | 97.59 | |
| | | BES 8 | 0.86 | 100.00 | |
| | QD-BES | | | | 0.75 |
| **Thailand** | Emotional satisfaction | BES 1 | 0.61 | 41.14 | 0.76 |
| | | BES 2 | 0.96 | 59.21 | |
| | | BES 3 | 0.96 | 69.65 | |
| | Support & respect by providers | BES 4 | 0.97 | 78.92 | 0.84 |
| | | BES 5 | 0.94 | 84.51 | |
| | | BES 9 | 0.58 | 89.48 | |
| | | BES 10 | 0.82 | 93.60 | |
| | Communication with providers | BES 6 | 0.50 | 96.18 | 0.82 |
| | | BES 7 | 0.88 | 98.46 | |
| | | BES 8 | 0.83 | 100.00 | |
| | QD-BES | | | | 0.85 |
| **Vietnam** | Emotional satisfaction | BES 1 | 0.60 | 35.93 | 0.74 |
| | | BES 2 | 0.78 | 52.77 | |
| | | BES 3 | 0.76 | 64.70 | |
| | Support & respect by providers | BES 4 | 0.87 | 73.54 | 0.86 |
| | | BES 5 | 0.91 | 80.03 | |
| | | BES 9 | 0.81 | 85.50 | |
| | | BES 10 | 0.70 | 90.27 | |
| | Communication with providers | BES 6 | 0.76 | 94.04 | 0.83 |
| | | BES 7 | 0.87 | 97.54 | |
| | | BES 8 | 0.88 | 100.00 | |
| | QD-BES | | | | 0.78 |

*Cronbach's alpha coefficient for the 10 items = 0.75.

**Table 3. Criterion-related validity results.**

| Hypothesis | Argen-tina | Burkina Faso | Thai-land | Viet Nam | Confirmed by statistical test | Dimension of QD-BES scale[1] |
|---|---|---|---|---|---|---|
| Women who preferred vaginal birth and gave birth vaginally are more satisfied than other women[2] | 0.037 | 0.001 | 0.002 | 0.006 | Yes | Emotional satisfaction |
| Women who breastfed within an hour of giving birth are more satisfied than those who started breastfeeding later or who did not breastfeed. | 0.031 | 0.001 | 0.001 | 0.001 | Yes | Emotional satisfaction |
| Primiparous women are more satisfied with childbirth care than multiparous women. | 0.253 | 0.63 | 0.56 | 0.09 | No | Support and respect by providers |
| Women who have had an episiotomy are less satisfied than women who have not had an episiotomy. | 0.001 | 0.049 | 0.033 | 0.032 | Yes | Support and respect by providers |
| Women who communicated with providers about their preferences are more satisfied compared to women with no communication with providers about preferences. | 0.015 | 0.001 | 0.001 | 0.038 | Yes | Communication with providers |
| Women with higher levels of education are more satisfied than women with less education. | 0.028 | 0.001 | 0.014 | 0.019 | Yes | Communication with providers |

[1]Dimension of the scale for which the association was statistically significant.

[2]Other women include 1) women who preferred vaginal birth and gave birth by caesarean section, 2) women who preferred caesarean section and gave birth vaginally, or 3) women who preferred caesarean section and gave birth by caesarean section).

Criterion-related validity testing was done by exploring several characteristics of women, their obstetric histories, and birth experiences using bivariate analysis. All hypotheses were tested and verified for each dimension and in each country; only one (primiparous women are more satisfied with childbirth care than multiparous women) was not verified for its dimension (support and respect by providers) (Table 3).

## Discussion

We present a psychometric validation of a scale measuring women's satisfaction with and experiences of care, using a scale that was constructed, developed, and validated in a LMIC using a systematic approach. Our 10-item scale is a short instrument, easily administered, valid, and reliable for measuring childbirth experience and satisfaction with childbirth care in four LMICs. This measurement scale showed satisfactory quality, reliability, and validity. The psychometric characteristics of the scale and dimensions are adequate based on acceptability, content validity, discriminant validity, reliability, and internal consistency measurement characteristics. The exploratory factor analysis favored a 3-factor solution (emotional satisfaction, support and respect by providers, and communication with providers), which was shown to explain over 67% of the variance. In particular, the QD-BES scale has high content validity based on our extensive use of the literature to inform item development and expert reviews.

In the QD-BES tool, we aimed to measure both satisfaction with care and experiences of care (particularly related to communication, respect, and emotional support), and acknowledged that these domains are closely interrelated. In the initial scale development, QD-BES items 1–5 were designed to measure satisfaction with care, whereas QD-BES items 6–10 were designed to measure experiences of care. In the 3-factor solution, we find that QD-BES items 1–3 loaded to 'emotional satisfaction', QD-BES items 4, 5, 9 and 10 loaded to 'support and respect by providers', and QD-BES items 6–8 loaded as 'communication with providers. This means that the QD-BES items did not load systematically into satisfaction versus experience conceptual categories, which is expected given the interrelation between the satisfaction and experience domains and related items. We note that for practical purposes of quality improvement, understanding which items on the QD-BES scale measure experiences of care (process measures) versus satisfaction with care (outcome measures) is important for interpretability, as experiences of care measures can be more sensitive to differences in quality of care and can be easier to make concrete actions to improve. In contrast, the satisfaction measures are interpreted as

assessments as to whether care met the person's needs and expectations, and may be useful to better understand areas of healthcare services that are important to individuals or communities.

Nilvér and colleague's adaptation of Terwee's quality criteria for measurement scales [10,50] includes consideration of the following properties: the need to develop and test a new instrument, face and content validity, internal consistency, criterion validity, construct validity, reproducibility agreement, reproducibility reliability, responsiveness, floor and ceiling effects, and interpretability. We have clearly justified the need for a new instrument based on the lack of validated scales in LMIC settings to measure birth experiences and satisfaction. In this analysis, we have described the measurement aim, target population, concepts to be measured and process for item selection (content validity). The factor analysis was performed on an adequate sample size, and the Cronbach's alpha for each dimension is > 0.70 (internal consistency). We developed six hypotheses regarding the directionality of relationships and five of our true, suggesting that the QD-BES scale is consistent with theoretically-derived concepts (construct validity). Our 3-factor solution can be easily and logically interpreted into the corresponding domains of emotional satisfaction, support and respect by providers, and communication with providers (interpretability). There is no gold standard tool to measure satisfaction or experience; thus criterion validity cannot be meaningfully assessed. While we do not have the data to assess reproducibility or responsiveness yet since the QUALI-DEC study is still in the intervention period, we will be able to assess reproducibility and responsiveness during the endline period.

Given the reliability and validity of the QD-BES scale, this scale can be validated in other LMIC settings to assess its appropriateness for use. The QD-BES scale is relatively short (10-items) and is therefore pragmatic and may be logistically easy to integrate into other implementation research in maternal health to ensure that women-reported measures are appropriately evaluated. Similarly, given the pragmatic nature of the scale, QD-BES may also be used as a simple measure for routine monitoring and evaluation of women's experiences and satisfaction with care for quality improvement. Quantitative evaluations of experiences of and satisfaction with maternity care should also be balanced with qualitative assessment, via in-depth interviews and/or focus group discussions with women and health workers. These qualitative approaches can help provide more nuanced understanding of women's perspectives, and may help elucidate how and why both positive and negative experiences happen.

### Limitations and strengths

As with any research, our study has some limitations. First, the test-retest reliability was not initially planned, meaning that assessment of the degree to which the woman's satisfaction scores are repeatable, i.e., how consistent their sum scores are across time, was not performed. Additionally, the study sample was taken from 32 hospitals, and therefore may not be of generalizability to other hospitals in the four study countries (noting that these study sites represent a mix of the types of hospitals in each country). Lastly, given that the surveys took place with women in the hospital setting, it is possible that woman may report responses more positively due to social desirability bias [51,52]. We aimed to mitigate this by employing social scientists who did not have a prior relationship with the hospitals and were not involved in patient care, to conduct the survey, and encourage the social scientists to reiterate to women that their responses would have no influence on their care and her confidentiality would be guaranteed.

Our study also has several strengths. First, the relatively large sample size over 3,000 women made it possible to randomly divide the participants into half subgroups, so that both an exploratory and a confirmatory factor analyses could be undertaken. Secondly, the response rate to the study was high, which may mitigate selection bias. Thirdly, the number of items in each subscale is appropriate: it is recommended that a minimum number of items per factor should be three [53,54]. Lastly, we used rigorous methods to develop and evaluate the QD-BES scale, based on systematic assessment of existing literature and extensive experience in conceptualizing and measuring these concepts. All ten QD-BES items had high loading coefficients, and we did not need to drop any items due to low-loading, thus suggesting that the systematic approach to tool development was rigorous.

## Conclusion

This paper reports the development of a new tool (QD-BES) to evaluate women's experiences of and satisfaction with childbirth care in LMIC and its validation in Argentina, Burkina Faso, Thailand and Viet Nam. Following a rigorous process of development, the scale to measure women's experience of and satisfaction with care has strong validity. We identified a 3-factor model to evaluate these domains consisting of: emotional satisfaction, support and respect by providers, and communication with providers, and these domains are well-grounded in theoretical and conceptual perspectives. This instrument will be useful to contribute to the generation of new knowledge about the quality of maternity care in LMICs, as well as help to meet the major challenge of implementing and measuring respectful care at scale.

## Supporting information

**S1 File. Inclusivity in global research.**
(DOCX)

## Acknowledgments

We thank all the data collectors, scientific and administrative staff of the participating institutions for their valuable contribution to this study. The QUALI-DEC research group is a consortium of researchers and implementers of nine institutions across Europe, Argentina, Burkina Faso, Thailand, and Vietnam. This group developed the QUALI-DEC project and is responsible for the implementation and the evaluation of the multifaceted intervention. The composition of the group is as follows: Karolinska Institutet (Sweden): Claudia Hanson, Helle Molsted-Alvesson, Kristi Sidney Annerstedt, Karen Zamboni; University College Dublin, National University of Ireland (Ireland): Michael Robson; World Health Organization (Switzerland): Ana Pilar Betràn, Newton Opiyo, Meghan Bohren; Centro Rosario de Estudios Perinatales (Argentina): Guillermo Carroli, Liana Campodonico, Celina Gialdini, Berenise Carroli, Gabriela Garcia Camacho, Daniel Giordano, Hugo Gamerro; CEDES (Argentina): Mariana Romero; Khon Kaen University (Thailand): Pisake Lumbiganon, Dittakarn Boriboonhirunsarn, Nampet Jampathong, Kiattisak Kongwattanakul, Ameporn Ratinthorn; Fundacio Blanquerna (Spain): Ramon Escuriet, Olga Canet; Centre national de recherche scientifique et technologique - Institut de Recherche en sciences de la sante (Burkina Faso): Charles Kabore, Yaya Bocoum Fadima, Simon Tiendrebeogo, Zerbo Roger; Pham Ngoc Thach University of Medicine (Vietnam): Mac Quoc Nhu Hung, Thao Truong, Tran Minh Thien Ngo, Bui Duc Toan, Huynh Nguyen Khanh Trang, Hoang Thi Diem Tuyet; Research Institute for Sustainable Development (France): Alexandre Dumont, Laurence Lombard, Myriam de Loenzien, Marion Ravit, Camille Etcheverry, Delia Visan.

## Author contributions

**Conceptualization:** Meghan A. Bohren, Alexandre Dumont, Ana Pilar Betrán.

**Data curation:** Charles Kabore, Kristi Sidney Annerstedt, Simon Tiendrebeogo, Fadima Bocoum, Marion Ravit, Pisake Lumbiganon, Nampet Jampathong, Guillermo Carroli, Celina Gialdini, Quoc Nhu Hung Mac.

**Formal analysis:** Meghan A. Bohren, Charbel Abi Saad, Kristi Sidney Annerstedt, Claudia Hanson, Myriam de Loenzien, Camille Etcheverry, Helle Mölsted Alvesson, Andrainolo Ravalihasy, Alexandre Dumont, Ana Pilar Betrán.

**Funding acquisition:** Meghan A. Bohren, Charles Kabore, Claudia Hanson, Myriam de Loenzien, Pisake Lumbiganon, Guillermo Carroli, Quoc Nhu Hung Mac, Alexandre Dumont, Ana Pilar Betrán.

**Methodology:** Meghan A. Bohren, Alexandre Dumont, Ana Pilar Betrán.

**Supervision:** Alexandre Dumont, Ana Pilar Betrán.

**Writing – original draft:** Meghan A. Bohren, Charbel Abi Saad.

**Writing – review & editing:** Meghan A. Bohren, Charles Kabore, Kristi Sidney Annerstedt, Claudia Hanson, Myriam de Loenzien, Simon Tiendrebeogo, Fadima Bocoum, Marion Ravit, Camille Etcheverry, Pisake Lumbiganon, Nampet Jampathong, Guillermo Carroli, Celina Gialdini, Quoc Nhu Hung Mac, Helle Mölsted Alvesson, Andrainolo Ravalihasy, Alexandre Dumont, Ana Pilar Betrán.

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
