## [Decision Letter · Decision Letter 0]

23 Dec 2024

PONE-D-24-32984Women’s experiences of and satisfaction with childbirth: development and validation of a measurement scale for low- and middle-income countriesPLOS ONE Dear Dr. Bohren,

Thank you for submitting your manuscript to PLOS ONE. After careful consideration, we feel that it has merit but does not fully meet PLOS ONE’s publication criteria as it currently stands. Therefore, we invite you to submit a revised version of the manuscript that addresses the points raised during the review process. In particular, kindly focus on the remarks concerning the statistical validation of the instrument.

We look forward to receiving your revised manuscript.

Kind regards,

Abiodun Adanikin, Ph.D

Academic Editor

PLOS ONE

“None declared.”

4. In this instance it seems there may be acceptable restrictions in place that prevent the public sharing of your minimal data. However, in line with our goal of ensuring long-term data availability to all interested researchers, PLOS’ Data Policy states that authors cannot be the sole named individuals responsible for ensuring data access (http://journals.plos.org/plosone/s/data-availability#loc-acceptable-data-sharing-methods).

Additional Editor Comments:

Please refer to reviewers' comments for revision.

Reviewers' comments:

Reviewer's Responses to Questions

**Comments to the Author**

1. Is the manuscript technically sound, and do the data support the conclusions?

Reviewer #1: Yes

Reviewer #2: Partly

2. Has the statistical analysis been performed appropriately and rigorously? 

Reviewer #1: I Don't Know

Reviewer #2: No

3. Have the authors made all data underlying the findings in their manuscript fully available?

Reviewer #1: Yes

Reviewer #2: Yes

4. Is the manuscript presented in an intelligible fashion and written in standard English?

Reviewer #1: Yes

Reviewer #2: Yes

5. Review Comments to the Author

Reviewer #1: This is a rigorous manuscript and makes a significant contribution to measuring person-centred outcomes in maternity care globally.

My main suggestion is around framing potential use of the tool. Given the broad nature of item questions, the authors could highlighting the importance of balance with qualitative feedback mechanisms within services, to understand the nuances of women’s and providers' experiences and what is required to improve services.

In the Abstract, it would seem important to mention the tool was developed in the context of improving decision-making for caesarean section and measuring program outcomes.

Please note: I can’t comment on the validity of the statistical methods as it is not my area of expertise.

Reviewer #2: Dear authors

Your effort in writing the article is clear and distinctive, and here are some points that need to be developed to make the research better, especially in the field of measurement and questionnaires and their validity.

There is a clear deficiency in the statistical tests used, and they are not sufficient

To judge the validity of the scale alone.

There are several statistical procedures that must be performed, including:

1.Cronbach's alpha

2.Guttman coefficient

3.Half-split coefficient

4.Principal Component Analysis

5.Exploratory and confirmatory factor analysis

Use the AMOS program to study these tests because it is the best in that

And it provides you with the charts that illustrate the necessary components of the scales you are developing

You should strongly review the steps for measuring validity used in this article as a comprehensive example

It is recommended to follow the same steps so that your work is at the highest scientific level of accuracy and comprehensiveness

https://link.springer.com/article/10.1186/s12909-024-05731-5

Finally, the method of calculating the sample size should also be explained in more detail.

6. PLOS authors have the option to publish the peer review history of their article (what does this mean? ). If published, this will include your full peer review and any attached files.

**Do you want your identity to be public for this peer review?** For information about this choice, including consent withdrawal, please see our Privacy Policy .

Reviewer #1: **Yes: ** Kayli Wild

Reviewer #2: **Yes: ** Imad-Addin Almasri

---

## [Author Response · Author response to Decision Letter 1]

27 Feb 2025

Please see the attached 'response to reviewers' document

---

## [Decision Letter · Decision Letter 1]

17 Mar 2025

Women’s experiences of and satisfaction with childbirth: development and validation of a measurement scale for low- and middle-income countries

PONE-D-24-32984R1

Dear Dr. Bohren,

We’re pleased to inform you that your manuscript has been judged scientifically suitable for publication and will be formally accepted for publication once it meets all outstanding technical requirements.

Kind regards,

Abiodun Adanikin, Ph.D

Academic Editor

PLOS ONE

Additional Editor Comments (optional):

Reviewers' comments:

Reviewer's Responses to Questions

**Comments to the Author**

1. If the authors have adequately addressed your comments raised in a previous round of review and you feel that this manuscript is now acceptable for publication, you may indicate that here to bypass the “Comments to the Author” section, enter your conflict of interest statement in the “Confidential to Editor” section, and submit your "Accept" recommendation.

Reviewer #1: All comments have been addressed

Reviewer #2: All comments have been addressed

2. Is the manuscript technically sound, and do the data support the conclusions?

Reviewer #1: Yes

Reviewer #2: Yes

3. Has the statistical analysis been performed appropriately and rigorously? 

Reviewer #1: I Don't Know

Reviewer #2: Yes

4. Have the authors made all data underlying the findings in their manuscript fully available?

Reviewer #1: Yes

Reviewer #2: Yes

5. Is the manuscript presented in an intelligible fashion and written in standard English?

Reviewer #1: Yes

Reviewer #2: Yes

6. Review Comments to the Author

Reviewer #1: The authors have addressed the two main issues I raised, around being more explicit about limitations of the tool.

Reviewer #2: Dear Authors,

Your Research is a valuable work that adds important updates to the medical literature.

Best wishes.

7. PLOS authors have the option to publish the peer review history of their article (what does this mean? ). If published, this will include your full peer review and any attached files.

**Do you want your identity to be public for this peer review?** For information about this choice, including consent withdrawal, please see our Privacy Policy .

Reviewer #1: **Yes: ** Kayli Wild

Reviewer #2: **Yes: ** Imad-Addin Almasri

---

## [Editor Report · Acceptance letter]

PONE-D-24-32984R1

PLOS ONE

Dear Dr. Bohren,

I'm pleased to inform you that your manuscript has been deemed suitable for publication in PLOS ONE. Congratulations! Your manuscript is now being handed over to our production team.

Kind regards,

on behalf of

Dr. Abiodun Adanikin

Academic Editor

PLOS ONE